# Transcriptomics and Functional Analysis of Copper Stress Response in the Sulfate-Reducing Bacterium *Desulfovibrio alaskensis* G20

**DOI:** 10.3390/ijms23031396

**Published:** 2022-01-26

**Authors:** Abhilash Kumar Tripathi, Priya Saxena, Payal Thakur, Shailabh Rauniyar, Dipayan Samanta, Vinoj Gopalakrishnan, Ram Nageena Singh, Rajesh Kumar Sani

**Affiliations:** 1Department of Chemical and Biological Engineering, South Dakota School of Mines and Technology, Rapid City, SD 57701, USA; abhilashkumar.tripathi@mines.sdsmt.edu (A.K.T.); priya.saxena@mines.sdsmt.edu (P.S.); payal.thakur@mines.sdsmt.edu (P.T.); shailabh.rauniyar@mines.sdsmt.edu (S.R.); dipayan.samanta@mines.sdsmt.edu (D.S.); vinoj.gopalakrishnan@sdsmt.edu (V.G.); Ram.Singh@sdsmt.edu (R.N.S.); 22-Dimensional Materials for Biofilm Engineering, Science and Technology, South Dakota School of Mines and Technology, Rapid City, SD 57701, USA; 3Data Driven Material Discovery Center for Bioengineering Innovation, South Dakota School of Mines and Technology, Rapid City, SD 57701, USA; 4BuG ReMeDEE Consortium, South Dakota School of Mines and Technology, Rapid City, SD 57701, USA; 5Composite and Nanocomposite Advanced Manufacturing Centre—Biomaterials, Rapid City, SD 57701, USA

**Keywords:** *Desulfovibrio alaskensis* G20, sulfate reducing bacteria, RNA sequencing, transcriptomics, gene expression, copper, toxicity, oxidative stress, homeostasis, gene ontology

## Abstract

Copper (Cu) is an essential micronutrient required as a co-factor in the catalytic center of many enzymes. However, excess Cu can generate pleiotropic effects in the microbial cell. In addition, leaching of Cu from pipelines results in elevated Cu concentration in the environment, which is of public health concern. Sulfate-reducing bacteria (SRB) have been demonstrated to grow in toxic levels of Cu. However, reports on Cu toxicity towards SRB have primarily focused on the degree of toxicity and subsequent elimination. Here, Cu(II) stress-related effects on a model SRB, *Desulfovibrio alaskensis* G20, is reported. Cu(II) stress effects were assessed as alterations in the transcriptome through RNA-Seq at varying Cu(II) concentrations (5 µM and 15 µM). In the pairwise comparison of control vs. 5 µM Cu(II), 61.43% of genes were downregulated, and 38.57% were upregulated. In control vs. 15 µM Cu(II), 49.51% of genes were downregulated, and 50.5% were upregulated. The results indicated that the expression of inorganic ion transporters and translation machinery was massively modulated. Moreover, changes in the expression of critical biological processes such as DNA transcription and signal transduction were observed at high Cu(II) concentrations. These results will help us better understand the Cu(II) stress-response mechanism and provide avenues for future research.

## 1. Introduction

Copper (Cu) is a naturally available element used as a micronutrient by bacteria and is also an essential co-factor for many redox-active enzymes [1]. However, Cu is also an antimicrobial agent and results in progressive inhibition of bacterial growth with increasing concentration [2]. Due to its widespread presence in aquatic systems, Cu is deemed a pollutant instead of being crucial for cell metabolism and survival [1,3]. Copper originates from various sources (highway runoff, sludge dumping, mine tailings, and industrial effluents) worldwide [4]. It is reported at elevated levels in ground water and soil [5]. In addition, Cu has been the metal of choice for piping that is used in household water delivery systems [6]. However, the leaching of Cu from such distribution systems is a source of various illnesses (vomiting, diarrhea, liver damage, and kidney disease) and, therefore, a public health concern [7]. Surplus Cu in the environment generates distinctive pleiotropic effects in the bacterial cell. Such pleiotropic effects generate varying levels of stress responses that complement the copper homeostasis system in protecting the cell against Cu toxicity [1]. The interest in microbe-based bioremediation of heavy metals has surfaced in recent years as scientists and engineers try to remove, recover, or stabilize heavy metals in soils and aquatic systems [8]. Previous studies have described Cu homeostasis mechanism in other bacteria such as *Escherichia coli, Pseudomonas aeruginosa,* and *Salmonella enterica* [9,10,11,12]. Studies have suggested that the general mechanism of Cu homeostasis can be considered common in the bacterial kingdom [1,9].

Bacteria handle Cu stress through well-regulated Cu homeostasis components both intracellularly and extracellularly. However, no single pathway can account for a cell’s Cu homeostasis profile, and several layers of gene regulation are needed towards Cu stress adaptation. Bacterial Cu homeostasis shows some distinctive characteristics where Cu exists in a chemical equilibrium between cupric (Cu(II)) and cuprous (Cu(I)) ions. These ions are readily complexed by various biological ligands as described in previous studies [4,13,14]. Cu(II) is present mainly in the oxidizing compartments (periplasm), while Cu(I) is prevalent in reducing compartments (cytoplasm) [1]. In general, Cu(II) is a relatively safe species and biologically more inert as compared to Cu(I) [9,11]. Cu can cross the bacterial outer membrane into the periplasm through Cu(I)/Cu(II) porin ATPases [1,15]. However, to date, only a few Cu transporters have been validated, and all of them have different structures and transport mechanisms [12].

Cu and other heavy-metal toxicity in the environment have been addressed previously using sulfate-reducing bacteria (SRB) [4,16,17,18]. SRB are heterotrophic anaerobic microorganisms that oxidize simple organic compounds by reducing sulfate to sulfide [19]. The sulfide thus generated can easily remove heavy metals such as Cu from contaminated sites by forming metal sulfide complexes [16]. However, the mechanism of Cu homeostasis in SRB is unknown. Reports concerning the toxicity of Cu and other heavy metals towards SRB have primarily focused on their degree of toxicity and subsequent elimination [4,20,21,22]. Considering the importance of Cu in various industrial settings and its subsequent release and interaction with SRB in the environment, it is crucial to understand the genetic and molecular level mechanisms of Cu stress tolerance in SRB. In bacteria (both Gram-positive and Gram-negative), a highly regulated system of transcriptional regulators, soluble chaperones, membrane transporters, and cupro-proteins regulates Cu homeostasis [10,23]. These biomolecules maintain beneficial levels of Cu that vary, corresponding to different bacteria. Considering the changes induced at the genetic and molecular level during Cu stress, it is imperative to study genome-wide transcriptional changes during Cu stress. Such transcriptomic studies correlate genomic information to biological function and molecular pathways, which helps identify key gene regulation targets related to growth and stress adaptation. To the best of our knowledge, there is no report on transcriptomic analysis of any SRB under inhibitory Cu(II) concentration.

This study was aimed at measuring the bacterial transcriptional response to the exposure to sublethal levels of Cu(II) ions in a model SRB, *Desulfovibrio alaskensis* G20 (DA-G20). RNA-sequencing technology was used to evaluate the expression pattern of multiple transcripts in DA-G20 exposed to Cu(II) ions. The bacterial response in terms of gene expression was further corelated to molecular functions and biological processes to understand the impact of Cu(II) stress on cellular pathways and molecular activities of gene products. The transcriptional response in DA-G20 was also compared with published transcriptional responses on other heavy metals to identify common and unique mechanisms of heavy metal stress tolerance. Overall, this study will enable the scientific community to obtain a more comprehensive picture of the biological processes impacted by Cu in other SRB species.

## 2. Results

### 2.1. DA-G20 Growth under Cu(II) Stress

To generate a baseline for Cu(II) toxicity in DA-G20, growth characteristics and changes in cell morphology were studied. The growth rate decreased as the concentration of Cu(II) increased (Figure 1a). The best-fitting linear model was a polynomial curve with the best goodness of fit (R^2^). The maximum specific growth rates (µ_max_) with 5 μM Cu(II) and 15 μM Cu(II) were found to be 0.13 and 0.10 h^−1^, respectively, compared to 0.17 h^−1^ for Cu(II) free control (0 μM Cu(II)). The trend in µ_max_ values indicates gradual inhibition in DA-G20 growth at high Cu(II) concentration. The growth profile of DA-G20 as a function of total cell protein was also studied in the presence of 0, 5, and 15 µM Cu(II) (Figure 1b). The results showed that with an increase in Cu(II) concentration, there is a significant decrease in total cell protein. It can be inferred from Figure 1b that the toxicity of Cu(II) was dependent on the Cu(II) concentration in the media. The final total cell protein in the presence of 5 μM and 15 μM Cu(II) was approximately 18.12% and 49.64% lower than Cu(II) free control, respectively. The inhibition of total cell protein at higher Cu(II) concentration resulted in longer lag times and lower growth rates, as discussed in a previous study [4]. As shown in Figure 1b, the cell protein concentration increased with time in 5 μM and 15 μM Cu(II) with almost similar lag times. The t_1/2_ (the time at which the total cell protein is half of the maximum total cell protein) was also calculated using a polynomial regression fit. The value of t_1/2_ increased with increasing Cu(II) concentration. The t_1/2_ for 0 μM Cu(II) (26.9 h) was closer to the t_1/2_ of 5 μM Cu(II) (29.6 h). However, at 15 μM Cu(II), the value of t_1/2_ increased to 52 h. In addition, scanning electron microscopy (SEM) was utilized to elucidate morphological changes in DA-G20 cells grown in varying concentrations of Cu(II) (0, 5, and 15 µM Cu(II)) (Figure 2). SEM analysis depicted a positive correlation between cell size and increasing Cu(II) concentration. The cell size increased as the concentration of Cu(II) was increased (Figure 2). Compared to 0 µM Cu(II) (2.538 µm), the cell size increased by 48% in 5 μM Cu(II) and 91.8% in 15 μM Cu(II). Additionally, spiral folds were also observed in cells grown in 5 μM Cu(II) and 15 μM Cu(II). The increase in cell size and presence of spiral folds might be due to pleiotropic effects generated by the presence of Cu(II) ions [24,25].

### 2.2. Differential Gene Expression Analysis during Cu(II) Stress

To explore potential molecular mechanisms of Cu(II) stress, global transcriptome analysis was performed on DA-G20 grown under varying concentrations of Cu(II) (0 µM, 5 µM, and 15 µM Cu(II)). Differential gene expression (DGE) analysis was performed on samples collected from the control (0 µM Cu(II)) and Cu(II) spiked samples (5 µM and 15 µM Cu(II)). DeSeq2′s median of ratios method was used for data normalization [26]. Pairwise transcriptome comparison between samples (SP1: 0 µM vs. 5 µM Cu(II); SP2: 0 µM vs. 15 µM Cu(II); SP3: 5 µM vs. 15 µM Cu(II)) was performed to identify differentially expressed genes (DEGs) amongst the sample pairs. From here on, the various pairwise transcriptome analysis is referred to as: SP1 for 0 µM vs. 5 µM Cu(II) comparison; SP2 for 0 µM vs. 15 µM Cu(II) comparison; SP3 for 5 µM vs. 15 µM Cu(II) comparison. DGE analysis after median of ratios normalization generated 1190, 1955, and 1144 genes with significant differential expression (*p*-value < 0.05 and |log_2_FC| > 0) in the pair-wise comparisons of SP1, SP2 and SP3, respectively (Figure 3a–c and Appendix A). In SP1, 61.43% of genes were downregulated, and 38.57% of genes were upregulated, whereas in SP2, 49.51% of genes were downregulated, and 50.5% of genes were upregulated. The pairwise comparison of both copper spiked samples (SP3: 5 µM vs. 15 µM Cu(II)) resulted in 52.97% downregulated and 47.03% upregulated genes. A Venn diagram was implemented to understand the association between DEGs in SP1, SP2, and SP3 (Appendix A). SP1 and SP2 shared a total of 1000 genes in which 373 are commonly upregulated, 613 are commonly downregulated, and 14 are contra-regulated. Around 18.6% DEGs were significant (*p*-value < 0.05) only at high Cu(II) concentration (SP2). Correlation-based on log_2_FC between the common genes in SP1 and SP2 shows a Pearson correlation coefficient of 0.87 (Figure 3d). A strong correlation exists between the expression of common genes in SP1 and SP2. In SP1, 61.43% of genes were downregulated, and 38.57% genes were upregulated (*p*-value < 0.05). Compared to SP1, 49.51% of genes were downregulated, and 50.5% of genes were upregulated in SP2. In SP3, 52.97% of genes were downregulated, and 47.02% were upregulated. A comprehensive table with protein names for all upregulated and downregulated genes in both SP1 and SP2 is given in Appendix A. These results suggest that most DEGs were observed at high Cu(II) concentration (SP2), which might be due to increased Cu stress in bacteria. Table 1 below lists the top 10 significantly up- and downregulated genes in SP1 and SP2.

The most downregulated gene (log_2_FC = −8.56) in SP2 was zinc resistance-associated protein (*ZraP* (Dde_0111)). ZraP is a zinc-responsive periplasmic protein that binds Cu(II) with higher affinity than Zn(II) [27]. The downregulation of *zraP* in the presence of 15 µM Cu(II) ions suggests a lower intake of Cu ions by DA-G20. *zraP* was also downregulated (log_2_FC = −2.42) in SP1, implicating a similar mechanism of Cu(II) stress tolerance. However, the most significantly upregulated (log_2_FC = 3.05) gene, serine/threonine phosphatase (Dde_3047) in SP1 was also upregulated (log_2_FC = 3.08) at the same level in SP2. In total, 373 genes were commonly upregulated between SP1 and SP2. However, the Pearson correlation coefficient (R) between these 373 genes was 0.3378, indicating a weak positive linear relationship between the shared upregulated genes. The R-value amongst the 613 commonly downregulated genes in SP1 and SP2 was 0.6017, indicating a more positive linear relationship. In addition, several significantly up- and downregulated uncharacterized proteins were also identified, which could play an important part in imparting Cu toxicity tolerance in DA-G20 (Table 1 and Appendix A). Finally, 15 genes were randomly selected for RT-qPCR validation of RNA-seq data. The expression pattern of the selected genes depicted the same expression trend as those detected by RNA-Seq, indicating good quality of the sequencing data (Appendix A).

### 2.3. Gene Ontology Analyses of DEGs

#### 2.3.1. Biological Process, Molecular Function, and Cellular Component Enrichment Analyses

Gene ontology (GO) terms approximating biological process (BP), molecular functions (MF), and cellular component (CC) were used for GO classification. BP represents pathways made up of the activities of multiple gene products; MF signifies molecular activities of gene products; CC represents the location of a protein relative to cellular compartments and structures. GO analysis was performed for all significantly upregulated and downregulated genes (*p*-value < 0.05) in SP1 and SP2. The five most significantly enriched terms (BP, MF, CC) on the basis of total gene count in both SP1 and SP2 are given in Table 2.

BP analysis revealed that the five most significantly enriched terms (*p*-value < 0.05) in SP1 were (i) regulation of transcription DNA-templated (GO:0006355), (ii) phosphorelay signal transduction system (GO:0000160), (iii) translation (GO:0006412), (iv) chemotaxis (GO:0006935), and (v) signal transduction (GO:0007165). Compared to SP1, the five most significantly enriched BP terms in SP2 were (i) translation (GO:0006412), (ii) regulation of transcription DNA-templated (GO:0006355), (iii) phosphorelay signal transduction system (GO:0000160), (iv) signal transduction (GO:0007165), and (v) methylation (GO:0032259). BP enrichment indicates that translation was significantly impacted when the concentration of Cu(II) was increased to 15 µM. Interestingly, methylation was also enriched in SP2, which implies methylation might play a crucial role in gene regulation during high Cu(II) stress in DA-G20 [28]. With regard to MF, the three most enriched terms in SP1 and SP2 were the same (ATP binding (GO:0005524), metal ion binding (GO:0046872), and hydrolase activity (GO:0016787)). The fourth and fifth most enriched terms in SP1 were 4 iron, 4 sulfur cluster binding (GO:0051539), and ATPase-coupled cation transmembrane transporter activity (GO:0022857), and 4 iron, 4 sulfur cluster binding (GO:0051539). The enrichment of metal ion binding (GO:0046872) in both samples (SP1 and SP2) suggests the involvement of molecular machinery involved in binding Cu(II) ions. In addition, the enrichment in ATP binding activity demonstrates that DA-G20 requires more energy in translocating heavy-metal ions using ATPase coupled transmembrane transporter activity. The top three CC enriched terms in SP1 and SP2 were integral component of membrane (GO:0016021), plasma membrane (GO:0005886), and cytoplasm (GO:0005737). The fourth and fifth most enriched terms in SP1 and SP2 were ATP-binding cassette (ABC) transporter complex (GO:0043190) and bacterial-type flagellum basal body (GO:0009425). The CC enrichment analysis touts the involvement of proteins belonging to permease, transporter, and various membrane-bound cytochrome families. These protein families help to maintain metal homeostasis in a diverse group of bacteria [29].

Bubble plots showing gene counts of all GO enriched terms in SP1 and SP2 are depicted in Figure 4a,b, respectively. Furthermore, the number of up- and downregulated genes for each GO enriched term is provided in Appendix A. Interestingly, the highest number of downregulated genes in both samples was associated with translation (GO:0006412), indicating lower turnover of protein during heavy-metal stress. Regulation of transcription, DNA-templated (GO:0006355), had more upregulated genes in SP2 (z-score = 1.5), whereas it had a similar number of upregulated and downregulated genes in SP1 (z-score = 0). With respect to CC, the most drastic difference between SP1 and SP2 was observed for integral component of membrane (GO:0016021). The z-score (Table 2) pertaining to GO:0016021 for SP1 was −0.41, whereas for SP2 it was 4.3. The overexpression of genes related to GO:0016021 in SP2 indicates that most transporter proteins were upregulated at high Cu(II) concentrations. Genes involved in ATP-binding cassette (ABC) transporter complex were overexpressed in both SP1 (z-score = 2.32) and SP2 (z-score = 4.12). The metal ion binding activity relatively increased in SP2 (z-score = −1.13) as compared to SP1 (z-score = −2.94). Interestingly, there was no difference in the expression level of chemotaxis-related genes in both SP1 and SP2 (z-score = −2.98). A comprehensive table showing gene names and the associated GO terms (BP and MF) for both SP1 and SP2 is given in Appendix A.

#### 2.3.2. Association between Genes and Enriched GO Terms

A comprehensive relationship between DGE and enriched GO functional categories (BP, MF) was interpreted by drawing a chord diagram using the GOChord function in R programming language [30]. The chord diagram for BP (Figure 5a,b) was drawn when at least 2 GO terms were assigned to a gene and at least 5 genes were assigned to a GO term. In the case of MF (Figure 5c,d), the chord diagram was drawn when at least 3 GO terms were assigned to a gene and at least 5 genes were assigned to at least one GO term. In SP1, the maximum number of GO-BP terms (3 each) was assigned to genes Dde_2411 and Dde_0703. Both of these genes (Dde_2411 and Dde_0703) code for proteins: methyl-accepting chemotaxis sensory transducer with Cache sensor and Pas/Pac sensor, respectively, which are part of bacterial two-component signal transduction system [31]. The most upregulated gene with 3 GO terms assigned in SP2 was Dde_0543, which codes for the lipid transporter protein-lipid II flippase (MurJ), that maintains cell shape and integrity during environmental stress [32]. Two more genes (Dde_0992 and Dde_1946) coding for peptidoglycan glycosyltransferase and undecaprenyl-diphosphatase, respectively, were assigned to 3 GO terms and are involved in cell-wall organization and peptidoglycan biosynthesis. The increased functional enrichment of genes involved in biological processes related to cell-wall maintenance highlights the importance of maintaining cell-wall integrity during heavy-metal stress in DA-G20. The gene with the most GO-MF terms assigned and the greatest fold change in SP1 was Dde_3105, which codes for iron-chelate-transporting ATPase (Figure 5c). This protein belongs to the class translocase and is involved in the translocation of inorganic cations. In SP2, Dde_0673 (glycine betaine/L-proline ABC transporter, ATPase subunit) was the most upregulated gene with three GO-MF terms assigned. In both SP1 and SP2, the most upregulated genes (Figure 5c,d) were primarily assigned to 3 GO-MF terms (ATP binding, hydrolase activity, and ATPase-coupled transmembrane transporter activity). The enrichment of these 3 GO terms indicates an enhanced need for energy in the form of ATP for the optimal functioning of enzymes involved in hydrolase and transmembrane transporter activity. Compared to SP2, the upregulation of genes associated with these 3 GO-MF enriched terms (ATP binding, hydrolase activity, and ATPase-coupled transmembrane transporter activity) was negligible (log_2_FC ≤ 1) in SP1, alluding to the fact that 15 µM Cu(II) was toxic enough to elicit metabolic responses in DA-G20 to maintain metal homeostasis. In addition, maintaining the homeostasis of transition metal ions such as Cu is essential for many physiological and biochemical processes required for bacterial metabolic activity [10]. The increase in expression of xenobiotic-transporting ATPase (Dde_1601) in SP2 (log_2_FC = 2.33) as compared to SP1 (log_2_FC = 0.52) indicates the ability of DA-G20 to maintain metal homeostasis at high Cu(II) concentration.

#### 2.3.3. Network Analysis of the DEGs and Identification of Clusters

STRING database was used to identify potential interactions between DEGs in SP1 and SP2. A protein-protein interaction (PPI) network was drawn for both SP1 and SP2, and 6 clusters of highly interconnected nodes were identified. Figure 6 shows all the significant clusters found in the PPI network analysis of samples SP1 and SP2. The most significant cluster in SP1 was related to stress response and chaperone activity, whereas in SP2, the most significant cluster was related to translation, RNA metabolic process, and tRNA processing. Most of the genes in cluster 1 were downregulated in both SP1 and SP2. The second most significant cluster in SP1 was related to flagellar assembly and chemotaxis, whereas in SP2, it was related to chaperone activity and stress response. Stress-response genes in SP2 were relatively upregulated as compared to SP1. Flagellar assembly and chemotaxis were combined into a single cluster (cluster 2) in SP1, whereas in SP2 flagellar assembly and chemotaxis were assigned into two different clusters (cluster 4 and cluster 5). However, a strong interaction between cluster 4 (flagellar assembly) and cluster 5 (chemotaxis) was seen in SP2, indicating co-regulation of genes involved in both processes. In SP1, the strongest inter-cluster correlation was between cluster 1 and cluster 4, indicating the impact of Cu(II) stress on the expression of the iron-sulfur binding domain-containing proteins. A strong correlation between cluster 4 and cluster 3 was also observed in SP2, demonstrating the impact of signal transduction genes on chemotaxis during Cu(II) stress. Individual StringDB score for every interaction in all the clusters in both SP1 and SP2 is given in Appendix A.

#### 2.3.4. Metabolomic Profile of Control and Test Samples

Targeted metabolomics analysis was carried out for all the samples (0 µM, 5 µM, and 15 µM Cu(II)). In total, 165 different intracellular and extracellular metabolites were identified in all the samples. Of these 165 metabolites, organic acids constituted 39.3% of the compounds, and amino acids constituted 20.6% of the compounds. The rest of the metabolites constituted nucleotides, alcohols, amines, and lipids. The intracellular metabolite with the highest concentration in media spiked with 5 µM and 15 µM Cu(II) was palmitic acid. Compared to control, the relative concentration of palmitic acid increased by approximately 7-fold and 3-fold in 5 µM and 15 µM Cu(II), respectively. In the extracellular environment, the metabolite with highest concentration was lactic acid, whose relative concentration increased by approximately 193-fold in 5 µM Cu(II) and 522-fold in 15 µM Cu(II). The increase in the concentration of lactic acid suggests that DA-G20 could not oxidize lactate in the presence of Cu(II) ions. Reduced lactate oxidation was also evident from downregulation (log_2_FC = −3.26) of lactate dehydrogenase (Dde_3239) in 15 µM Cu(II). An increase in concentration of some specific amino acids was also observed in 5 µM and 15 µM Cu(II) as compared to control. For example, in the extracellular environment, the amino acid with the highest concentration in 15 µM Cu(II) was pyroglutamic acid, while the intracellular amino acid with the highest concentration was alanine. However, the most interesting observation was the presence of ascorbic acid due to its involvement in the reduction of Cu(II) to Cu(I). The relative concentration of ascorbic acid increased with an increase in the concentration of Cu(II). Ascorbic acid was approximately 8-fold more in the presence of 15 µM Cu(II) as compared to control. Appendix A shows the concentration of all intracellular and extracellular metabolites observed in the presence of 0 µM, 5 µM, and 15 µM Cu(II).

## 3. Discussion

Bacteria rapidly evolve countermeasures to resist heavy-metal stress through changes in gene expression and synthesis of specific molecules that alleviate toxicity [33,34]. While numerous studies have been conducted on understanding gene-level mechanisms towards heavy-metal stress in other bacteria, this study focused on a specific SRB (DA-G20). SRB are a diverse group of anaerobic bacteria that play an important role in the global cycling of carbon and sulfur [35]. In this study, RNA-Seq and metabolomics analysis was used to investigate the effect of Cu(II) stress on DA-G20 growth and differential gene expression. The key finding of this study is the establishment of transcriptional and metabolite alteration that occurs in DA-G20 in response to Cu(II) stress and the survival strategies used by DA-G20 in the presence of Cu ions. The findings indicated that several Cu-related stress effects in DA-G20 were comparable to the effects of other heavy-metal associated oxidative stress in many other groups of Gram-negative bacteria [36,37,38,39]. However, there were some unique effects that were not reported in previous studies conducted on heavy metals. For instance, the upregulation of ApbE family lipoprotein (log_2_FC = 3.95) and modulation of methylation were not reported in past studies. DGE analysis showed that as compared to 5 µM Cu(II), media spiked with 15 µM Cu(II) had more DEGs. This indicates the ability of DA-G20 to adapt to increasing Cu(II) concentration by regulating gene expression. The data also revealed some uncharacterized DEGs that were significantly regulated in both SP1 and SP2. These uncharacterized proteins need to be classified as they probably play important roles in Cu stress tolerance in DA-G20. Some of the common transcriptional changes induced by Cu(II) ions are discussed in the following sections.

### 3.1. Downregulation of Translation Machinery

Most of the cellular ATP is used in the synthesis of proteins to carry out various biological processes and molecular functions [40]. The total downregulation of translation machinery in both SP1 and SP2 indicates that DA-G20 is saving most energy for stress response rather than translation. The same method of energy conservation was observed in previous studies conducted on other heavy metals. For instance, Joudeh et al. [36] reported that in the presence of palladium (1 mM), there was complete downregulation of translation apparatus, including the expression of r-proteins and tRNA processing genes in *E. coli* K-12. Similarly, translation arrest and energy conservation were also reported in the presence of other heavy metals such as cadmium [41], nickel [42], and silver [43]. Most of the downregulated genes in translation encode for 30S and 50S ribosomal protein in both SP1 (92.5% downregulated) and SP2 (96.3% downregulated). The small subunit (30S) decodes genetic information delivered by mRNA, whereas the larger subunit (50S) synthesizes polypeptides from amino acids delivered by tRNAs [44]. This further corroborates that in the presence of Cu ions, DA-G20 cells save energy at the expense of translation. Translation was more suppressed in the presence of 15 µM Cu (z-score = −7.07) than in the presence of 5 µM Cu (z-score = −4.81), possibly due to more oxidative stress in SP2 (Table 2).

### 3.2. Modulation in Transporter Related Activity

Transporters are essential for bacterial survival and growth as they are not only involved in the uptake of nutrients but also participate in defense mechanisms against environmental stress [45]. The most upregulated family of transporters in both SP1 and SP2 belonged to the ATP-Binding Cassette (ABC) superfamily of transporters. ABC is the largest protein transporter superfamily, and it consists of several proteins that are capable of transporting various inorganic and organic solutes across membranes [46,47]. The most upregulated ABC type transporter in SP2 was Dde_0611 (branched-chain amino acid transport system permease protein (LivH)). In addition, other ABC-type amino acid transporters (Dde_0673, Dde_3743, Dde_1430) were also upregulated in SP2. These genes code for glycine/betaine/L-proline ABC transporter (Dde_0673) and polar amino acid ABC transporter (Dde_3743 and Dde_1430). The same amino acid ABC transporters (except Dde_1430) were also upregulated in SP1. The overexpression of ABC-type amino acid transporters suggests an osmotic upshift in the presence of Cu ions. Betaine and proline are osmotolerant molecules, and the upregulation of Dde_0673 (Glycine betaine/L-proline ABC transporter, ATPase subunit) could be one of the mechanisms of Cu stress regulation and maintenance of osmotic balance in DA-G20 [48,49]. The overexpression of these amino acid transporters may possibly help the cell to reverse the osmotic shrinkage of the cell and maintain cellular integrity [50]. However, certain ABC-type transporters (tungstate and molybdenum ABC transporter) were downregulated in both SP1 and SP2. The downregulation of tungstate and molybdenum ABC transporter was probably due to the downregulation of genes coding for proteins that require molybdenum and tungstate as co-factors. Interestingly, all ferrous iron (Fe(II)) transporter systems were downregulated in both SP1 and SP2. This clearly implies the disruption in Fe homeostasis in the presence of Cu and indicates that these systems could be more directly involved in Cu transport to the cell. The same downregulation trend was observed for Fe transporters in studies on other heavy-metal stress, including palladium [36], cobalt [51], mercury [52], and nickel [42].

In addition to ABC type transporters, transporters belonging to the resistance-nodulation-cell division (RND)superfamily and heavy-metal translocating P-type ATPases were also modulated in both SP1 and SP2. ATPase-coupled transmembrane transporter proteins such as ATP-binding cassette (ABC) transporters and P-type ATPases regulate the rate and mechanism of metal uptake and efflux [53]. The RND superfamily consists of seven protein families that are involved in the transport of heavy metals and hydrophobic compounds [54]. In this study, the RND efflux system genes were downregulated in both SP1 and SP2. Similarly, the heavy-metal translocating P-type ATPase genes were also downregulated. This observation is counterintuitive to results obtained on other heavy metals (cadmium and zinc) where the heavy-metal translocating P-type ATPases and heavy-metal RND genes were upregulated [55,56]. The downregulation of heavy-metal transporting ATPase was probably to maintain cellular homeostasis of Cu ions. One more plausible reason for this might be that metal ions antagonize metal uptake by inappropriately signaling sufficiency to transporters that are regulated allosterically [57]. A more detailed time-point study is required to elucidate temporal changes in the expression of these proteins. A previous study on Cu toxicity assessment in *P. aeruginosa* Q9I147 observed that when bacterial cells attained an optimum level of intracellular Cu ions required for metabolic needs, the heavy-metal transporting ATPase was downregulated [58]. The downregulation of the RND efflux system in DA-G20 shows that it is probably highly sensitive to Cu ions. Similar results were observed for the RND efflux system, where its expression was induced in the presence of cadmium or zinc but not in the presence of cobalt [56]. However, for certain bacteria, Cu enters the cell via passive diffusion along its chemical gradient [1]. There is a possibility that Cu ions enter DA-G20 through passive diffusion since most of the heavy-metal translocating ATPases are downregulated.

### 3.3. Regulation of Oxidative Stress Response

The results suggest an upregulation in the signal transduction pathways in SP2 as compared to SP1. The upregulation in signal transduction pathways might be due to increased oxidative stress in DA-G20 in the presence of high Cu(II) concentration. Metal-induced oxidative stress changes the redox state of the bacterial cell and impacts their ability to self-oxidation [36]. Generally, the reactive intermediates (mainly radicals) react with oxygen to form reactive oxygen species (ROS) such as superoxide anion (O_2_^−^) and hydrogen peroxide (H_2_O_2_) [59]. However, ROS may also be generated by autooxidation of dehydrogenases, thiols, flavins, and oxidases [60]. ROS disrupts normal cellular function as they are implicated in severe damage to nucleic acids, lipids, and proteins [61]. In this study, upregulation in signal transduction pathways at high Cu(II) concentration was observed. Upregulation in signaling pathways was also observed in a study on the assessment of palladium toxicity in *E. coli* [36]. Apart from modulating the expression of signaling pathways, cytosolic buffering is another process through which bacteria limit oxidative damage caused by the sudden influx of metal ions [62]. In general, the cellular metal buffering system consists of small molecules (amino acids, glutathione, organic acids, inorganic ligands and weak ligands) on the surface of specific buffering proteins and a subset of delivery proteins [57]. GC-MS metabolite analysis showed increased concentration of some specific organic acids (2-Keto-L-gluconic acid, citric acid, glyceric acid, lactic acid, 2-methyl-2,3-dihydroxy propanoic acid, succinic acid, stearic acid) in the supernatant. The increased concentration of these organic acids in 15 µM Cu(II) implies inhibition in metabolization of these acids. The increased concentration of organic acids also indicates that they might play a role in the cellular metal buffering system. However, specific components of cellular metal buffering are starting to be elucidated and more research on the role of organic acids in cellular metal buffering is required. In addition to organic acids, bacteria use specific amino acids to prevent oxidative stress through cellular metal buffering [57]. The relative concentration of methionine significantly increased in the presence of 5 µM Cu(II) and 15 µM Cu(II). Methionine is a sulfur-containing amino acid that forms the metal binding motif of many proteins and is involved in Cu(II) ion binding [63]. Many upregulated genes (including uncharacterized) in this study might code for proteins that have a methionine-rich motif and could be involved in Cu(II) ion binding. Similarly, a very high concentration of glycine, leucine, and isoleucine in the presence of 15 µM Cu(II) indicates upregulation of specific amino acid biosynthetic pathways during Cu(II) stress. However, in contrast to other studies, certain enzymes involved in removal of superoxide radicals were downregulated at high Cu(II) concentration (SP2). For example, superoxide dismutase (Dde_0882), thiol peroxidase (Dde_2313), and thioredoxin peroxidase were downregulated in the presence of 15 µM Cu(II). The genes coding for these enzymes were upregulated in studies on other heavy metals such as palladium and zinc [36,64]. These counterintuitive results suggests that DA-G20 might be using a completely different mechanism to counteract Cu(II) induced oxidative stress, and more research is required to elucidate the oxidative stress-tolerance mechanism in DA-G20. A starting point for deciphering the stress-response mechanism could be to check the activity of hydrolases. Hydrolase activity was significantly upregulated in the presence of 15 µM Cu(II) as compared to 5 µM Cu(II). Studies have reported increased hydrolase activity and more integrated action of hydrolases during heavy-metal stress [65,66,67]. The upregulation in hydrolase activity in this study warrants more investigation into the role of hydrolases during Cu toxicity in DA-G20.

### 3.4. Impact on Chemotaxis and Signal Transduction System

Chemotaxis is a process by which microorganisms sense and respond to changes in their environment through regulation in gene expression and/or active movement towards or away from a stimulus [68]. Bacterial chemotaxis involves chemosensory pathways, which are part of the two-component system (TCS) pathway signal transduction system [33]. In bacteria, the chemotaxis system consists of methyl-accepting chemotaxis proteins (MCPs), cytosolic chemotaxis proteins (Che proteins), and flagella. In this study, many genes coding for Che proteins (*cheD*, *cheW*, *cheC*, *cheZ*) involved in the regulation of the signal transduction system were downregulated in both SP1 and SP2. However, the signal transduction pathways were comparatively more upregulated in SP2 as compared to SP1 (Table 2). In SP1, Dde_2411 and Dde_0703 were assigned the most GO-BP terms (Figure 5). Dde_2411 and Dde_0703 code for proteins that are involved in monitoring cation accumulation and regulating the expression of related metabolic responses to changes in the concentration of heavy-metal ions [69]. Chemotaxis-related proteins also regulate bacterial motility by regulating flagella behavior. The expression of flagella is regulated by the gradient of attractant or repellant substrate. When the cells sense an increase in concentration of attractants, the expression of flagella increases, and bacteria swim for longer times, whereas when they sense more repellants, the expression of flagella decreases [70]. Many flagella-related proteins were downregulated in both SP1 and SP2, indicating that Cu(II) acts as a heavy-metal repellant in DA-G20. The same phenomena of downregulation in cell motility-related proteins were observed in studies conducted on other heavy metals [71,72]. This indicates that reduction in cell motility and chemotaxis is a widespread mechanism in bacterial heavy-metal stress response. In addition to cell motility, another important component of the signal transduction pathway are the sigma (σ) factors that provide tolerance towards heavy-metal stress by regulating the expression of outer membrane porins [73]. In this study, the gene (Dde_3097), which codes for Per-ARNT-Sim (PAS) domain σ^54^ transcriptional regulator, was upregulated at high Cu(II) concentration (SP2). The upregulation in the phosphorelay signal transduction system facilitates activation of σ^54^ through phosphorylation by a sensor-kinase protein [33]. In addition to the σ^54^ transcriptional regulator, the RNA polymerase sigma factor (σ^70^) was also upregulated in SP2; σ^70^ is involved in the transcription of genes that are essential for bacterial metabolism, growth, and stress responses [33]. The upregulation of σ^70^ might be an adaptive response in DA-G20 to counteract Cu(II) toxicity and maintain optimal cellular metabolism and growth. Additionally, serine/threonine phosphatase (Dde_3047), which belongs to the two-component signaling system (TCS) was upregulated in both SP1 and SP2 at the same level (log_2_FC ≈ 3). TCS is capable of phosphorylating response regulator (RR) proteins to elicit appropriate cellular response in response to an environmental stimuli [74].

In summary, it was observed that Cu(II) stress response in DA-G20 shares certain commonalities when compared to other heavy-metal stress in different bacteria. Substantial difference demonstrated by differential regulation in certain biological processes and molecular functions related to stress response and transporter activity was also observed. However, certain unique transcriptional patterns were observed, which warrants further investigation into understanding Cu stress response in DA-G20. Some of these unique changes are discussed in brief in the following sections.

### 3.5. Some Atypical Transcriptional Changes Induced by Cu(II) Ions

Some transcriptional changes detected in this study were not observed or discussed in previous studies on heavy-metal stress. This section summarizes findings on such uncommon transcriptional changes. This section also discusses some counterintuitive results compared to other heavy-metal stress studies.

#### 3.5.1. Significant Upregulation of ApbE Family Lipoprotein

The most upregulated gene (15-fold) in SP2 was Dde_2535 (ApbE family lipoprotein). ApbE family lipoprotein is part of the ApbE superfamily and is a periplasmic lipoprotein anchored to the inner membrane [75]. The periplasmic ApbE lipoprotein has been implicated in thiamine biosynthesis [76], iron-sulfur cluster metabolism [77], and flavin transferase activity [78]. Interestingly, Dde_2535 was not significantly expressed (*p*-value > 0.05) in SP1 (5 µM Cu(II)), implying that its activation was only triggered at high Cu(II) concentration. Dde_2535, therefore, might play an important role in Cu(II) stress tolerance in DA-G20. However, ApbE proteins are relatively understudied biochemically, and more targeted studies involving gene-knockout and Cu(II) stress response in the mutant could explain more about the role of ApbE proteins in Cu(II) tolerance in DA-G20.

#### 3.5.2. Differential Expression of DNA Repair Genes

Heavy-metal-induced oxidative stress generates ROS, causing damage to many cellular components, including DNA and proteins [37]. In this study, eleven DNA repair enzymes were upregulated, and six were downregulated at high (15 µM) Cu(II) concentration (SP2). At low (5 µM) Cu(II) concentration (SP2), seven DNA repair enzymes were upregulated, and five were downregulated (Appendix A). Previous studies have observed modulation in the expression of DNA repair genes in the presence of specific heavy metals. For example, upregulation in DNA repair genes was observed in the case of mercury, cadmium, zinc, and arsenic [52,65,79,80]. However, palladium exposure resulted in the downregulation of DNA repair genes [36], and no change in the expression pattern of DNA repair genes was observed in the case of nickel, copper, and cobalt [39,42,51]. In this study, specific DNA repair genes were both upregulated and downregulated at 15 µM Cu(II) concentration. Dde_2324 was 3.5-fold (log_2_FC = 1.83) upregulated in SP2 as compared to SP1. Dde_2324 codes for the protein resolvase (*ruvC*), which is a specialized nuclease that removes Holliday junctions (HJs) during DNA repair [81]. HJs are cross-stranded four-way junction structures formed during homologous recombination. These structures are detrimental to cell viability and must be removed in a timely manner to maintain genetic stability [82]. Modulation in the expression of *ruvC* to the best of our knowledge was not reported before in any heavy-metal stress study. The upregulation of *ruvC* in SP2 suggests that removal of HJs is an essential component Cu(II) stress tolerance mechanism in DA-G20. However, more targeted transcriptomics studies, including gene knockout, are required to establish the exact role of *ruvC* and other DNA repair genes during heavy-metal stress.

#### 3.5.3. Modulation of Methylation Related Activity

Methylation in bacteria is used as a signal for the epigenetic control of DNA-protein interactions that control gene regulation [83]. DNA methylation also protects genome integrity in prokaryotes [84]. This study showed an upregulation of ten methyltransferase coding genes at high (15 µM) Cu(II) concentration. In contrast, at low (5 µM) Cu(II) concentration, only one methyltransferase gene was upregulated. Upregulation of methyltransferases at high Cu(II) concentration indicates that methylation probably changes expression patterns of daughter cells based on environmental conditions [85]. The most upregulated methylation associated gene was Dde_2875 (methyltransferase FkbM family). To the best of our knowledge, modulation in the genes related to methylation was not reported before in any heavy metal or oxidative stress study. The reason for such modulation is probably related to gene regulation associated with environmental adaptations. However, the exact mechanism of regulation is not known and provides avenues for future research.

#### 3.5.4. Regulation of Genes Associated with Cell Division and Cell Wall Organization

Some genes involved in biological processes such as cell division, cell-wall organization, and cell cycle appeared to be both downregulated and upregulated. In SP2, 24 genes associated with these biological processes were differentially expressed, whereas in SP1, only 12 were differentially regulated. The most upregulated gene (log_2_FC = 2.11) in SP2 (15 µM Cu(II)) was Dde_0543 (lipid II flippase, MurJ). In comparison, Dde_0543 was only slightly modulated (log_2_FC = 0.68) in SP1 (5 µM Cu(II)). Lipid II flippase is essential in the regulation of cell shape and maintenance of cellular integrity during environmental stress. Depletion of MurJ causes cells to swell and burst, and as such, the overexpression of gene coding this protein seems to be essential during Cu(II) stress in DA-G20 [86]. An intriguing observation was the downregulation of an uncharacterized protein (Dde_0980) in both SP1 (log_2_FC = −1.78) and SP2 (log_2_FC = −2.79). The GO analysis indicated that this uncharacterized protein is involved in two biological processes: (i) cell septum assembly and (ii) FtsZ-dependent cytokinesis. FtsZ is a major cytoskeletal protein that initiates cell division in a bacterial cell [87]. Previous studies have shown that downregulation in *ftsZ* expression results in a transient delay in cell division that leads to an increase in cell size [88,89]. Another captivating observation was the downregulation of Dde_2171 in both SP1 (log_2_FC = −1.88) and SP2 (log_2_FC = −2.52). Dde_2171 codes for DivIVA domain-containing protein, which is a cell division initiation protein. Stationary phase or slow-growing cells have DivIVA at both cell poles; however, actively growing cells reduce the DivIVA concentration at the old poles and redeploy DivIVA molecules to build DivIVA rings at active division sites [90]. The downregulation of the *DivIVA* gene is probably an indication of slow growth and an increase in cell size at high Cu(II) concentration. In addition, the downregulation of *divIVA* might be explained by the poor health status of the bacterial cells that favor the downregulation of these genes in order to conserve energy for stress-related pathways. However, literature search showed that at present, there are no studies on the impact of heavy-metal stress on the DivIVA domain-containing protein. As such, the impact of heavy-metal stress on cell cycle, cell division, and cell shape is still poorly understood, and more research is required to better understand this impact. The impact of *ftsZ* and *divIVA* genes on cell size seems apparent; however, there is a possibility that mutations in some uncharacterized genes are responsible for generating pleiotropic effects that influence cell morphological characteristics, including cell size [25]. These observations suggest the need of genome-wide association studies that will shed more light on the contribution of some specific genes on observable phenotypic traits.

#### 3.5.5. Role of Ascorbic Acid and Alkaline Phosphatase

The most important group of transporters during heavy-metal stress are copper influx and efflux proteins. Specific protein machineries are involved in fine-tuning the balance of intracellular copper trafficking and extracellular secretion according to cellular requirement [12]. However, the form of Cu ion (Cu(I) or Cu(II)) and the type of protein-import system involved in copper trafficking inside the cell are debatable. For instance, Andrei et al. [23] implicated that the outer membrane porin OmpF is involved in the influx of Cu(II), whereas Giachino and Waldron [1] proposed that OmpF is involved in the influx of Cu(I) ions. In this study, it was observed that an increase in the concentration of ascorbic acid in the supernatant occurred when Cu(II) concentrations increased.Ascorbic acid is reported to be involved in the conversion of Cu(II) to Cu(I) [91]. The increase in expression of alkaline phosphatase (ALK) (Dde_1612) also corroborated the role of ascorbic acid in Cu(II) reduction as ALK hydrolyzes 2-phospho-Lascorbicacid (AAP) to generate ascorbic acid (AA) in the extracellular environment [91]. Cu(I) is relatively more toxic compared to Cu(II), and the reduction of Cu(II) to Cu(I) by ascorbic acid probably inhibits DA-G20 growth [1]. However, the mechanism of Cu(I) transport and growth inhibition in the presence of Cu(I) is still not clear. In addition to heavy-metal translocating P-type ATPases, Cu can also enter the cell via passive diffusion along its chemical gradient [1]. Based on the analysis of the metabolomics and RNA-seq data, a putative mechanism of Cu stress response in DA-G20 was devised and is presented in Figure 7.

## 4. Materials and Methods

### 4.1. Bacterial Strain and Growth Conditions

The bacterial strain used in this study was *Desulfovibrio alaskensis* strain G20 (DA-G20). Lactate C medium (anoxic) was used to grow DA-G20 at 30 °C and 125 rpm. The composition of the media is given in Appendix A. Medium (100 mL) was prepared in serum bottles and sterilized by autoclaving at 121 °C for 15 min. The medium was then deoxygenated by sparging filter-sterilized ultrapure nitrogen (10 psi) for 20 min as described previously [4,18].

### 4.2. Copper Toxicity Experimental Setup

Seed culture of strain DA-G20 was prepared by inoculating 1.5 mL of a frozen stock (40% *v*/*v* glycerol stock solution and cells grown in lactate-C medium) into 100 mL of lactate-C medium. The seed culture was incubated at 30 °C with an initial pH of 7.2 and 125 rpm. Hydrogen sulfide (H_2_S) initially present in the active inoculum was purged out using filter-sterilized ultrapure nitrogen for 1 h. From this sparged culture, 5% *v*/*v* active inoculum (OD = 0.18) was used for Cu(II) toxicity experiments. For Cu(II) toxicity experiments, filter-sterilized stock solution of 0.05 M CuCl_2_ was prepared and supplemented to serum bottles containing 100 mL lactate-C medium to give the desired metal concentrations (5 µM and 15 µM). CuCl_2_ dissociates into Cu(II) ions, and the Cu toxicity mentioned in this study refers to toxicity by Cu(II) ions. The maximum concentration was selected as 15 µM Cu(II) by preliminary experiments, since above this concentration, the growth rate reduced significantly, and the recovered RNA quantity was not sufficient for transcriptomic studies. Another reason for selecting 15 µM Cu(II) as the maximum concentration was approximately 50% inhibition in total cell protein at this concentration. The serum bottle containing no supplemented Cu(II) (0 µM) was used as control for the experiment. All the experiments were carried out in triplicates for statistical validation.

### 4.3. Determination of Total Cell Protein and SEM Analysis

The total cell protein was estimated for each experimental condition (0 µM Cu(II), 5 µM Cu(II), 15 µM Cu(II)) using a quantitative Coomassie (Bradford) protein assay kit (Thermo Fisher Scientific, Waltham, MA, USA). Protein concentrations were calculated by comparing the absorbance (595 nm) of all samples (0 µM, 5 µM, 15 µM Cu(II)) to the color response of protein assay standards prepared as a series of known dilutions of bovine serum albumin (BSA). Polynomial regression was used to estimate the time at which total cell protein was half of the maximum cell protein. All the fit curves had R^2^ greater than 0.95. The change in cell morphology (planktonic cells) was visualized using a field-emission scanning electron microscope (FESEM (Zeiss Supra40 variable pressure)). The samples for SEM were fixed by immersing in a mixture of 0.1 M sodium cacodylate buffer and 2% glutaraldehyde overnight at 4 °C and then at room temperature for 1 h. It was followed by sequential dehydration with ethanol (50%, 75%, 95%, and 100%) for 30 min each, followed by overnight drying in a desiccator. The SEM images were obtained using secondary electron imaging operated at an accelerated voltage of 1 kV.

### 4.4. RNA Isolation

Total RNA was extracted from 10 mL of bacterial culture from the serum bottles on the 5th day from different Cu(II) concentrations. To pellet down the cells, 10 mL of the DA-G20 culture was taken out from the respective serum bottles and was centrifuged at 10,000× *g* for 10 min at 4 °C. The cell pellet was collected and washed three times with anaerobic phosphate buffer saline (pH 7.2) to remove any unwanted media components that may hamper RNA extraction. Cell pellet was then transferred to sterile 2 mL microcentrifuge tubes for further extraction steps. Total RNA was extracted using the TRIzol RNA extraction method and eluted in 20 µL nuclease-free water [92]. Extracted RNA was further purified using the RNA clean and concentrator kit following the manufacturer’s instructions (Zymo Research, Irvine, CA, USA) and eluted in 10 µL nuclease-free water. RNA purity was assessed using Nanodrop UV-Vis spectrophotometer (Thermo Fisher Scientific, Waltham, MA, USA). The concentration was further evaluated using the Qubit RNA assay kit and Qubit 3.0 fluorometer (Thermo Fisher Scientific, Waltham, MA, USA). Finally, RNA integrity was assessed using the Bioanalyzer 2100 system (Agilent Technologies, Santa Clara, CA, USA).

### 4.5. Complementary DNA (cDNA) Library Preparation and Sequencing

Library preparation and RNA sequencing were performed by Oklahoma Medical Research Foundation NGS Core (Oklahoma city, OK, USA). The protocol contained the following steps: First, rRNA was depleted using RiboCop rRNA Depletion Kit (Lexogen, Greenland, NH, USA). cDNA library was then prepared using Swift Rapid RNA Library kit using the manufacturer’s protocol (Swift Biosciences, Ann Arbor, MI, USA). The final quality control of the prepared library was done using Kapa qPCR and Agilent Tapestation 4150 (Agilent Technologies, Santa Clara, CA, USA). The library was sequenced using the Illumina-NovaSeq platform using an S4 flow cell with a 150-bp paired-end module. Approximately 20 million reads (10 million in each direction) were generated per sample. All the samples were used as biological triplicates for statistical validation.

### 4.6. QC of Raw RNA Sequencing Reads and Data Analysis

To ensure the quality of subsequent data analysis, the raw RNA-seq reads were subjected to quality control using FASTQC tool that generated average Q scores across the length of all sequence files [93]. The data obtained after initial QC was processed using Trimmomatic (version 0.38) [94]. An illuminaclip step was used to remove adapter sequences (TruSeq3 pair ended) from the reads. The raw reads were further filtered using a sliding window filtration to truncate reads at a 4-base average Q score of 20 or lower. The sequences greater than 20 bp were retained. The QC reads were then mapped against DA-G20 reference genome (accession PRJNA329) using the HISAT2 alignment program [95]. The number of reads mapping to each gene was counted using DA-G20 genome annotation file (GFF3, Ensembl) and read summarization program: featureCounts [96]. The gene expression fold change was estimated amongst various sample groups (0 µM vs. 5 µM Cu(II), 0 µM vs. 15 µM Cu(II), 5 µM vs. 15 µM Cu(II)) using the DESeq2 package [26]. DeSeq2 uses the median of ratios normalization for differential gene expression analysis between samples. The data analysis (until DESeq2) was done using the web-based scientific analysis platform Galaxy [97].

### 4.7. RT-qPCR Validation

Some differentially expressed genes were randomly selected for RT-qPCR analysis to validate the quality of sequencing data. RNA was extracted and purified as described in the previous section on RNA isolation. cDNA synthesis was performed using the QuantiTect Reverse Transcription Kit (Qiagen, Germantown, MD, USA). Subsequently, RT-qPCR was performed in a QuantStudio 3 real-time PCR system (Thermo Fisher Scientific, Waltham, MA, USA) using the Maxima SYBR Green/ROX qPCR Master Mix (Thermo Fisher Scientific, Waltham, MA, USA) in a 96-well plate. Gene-specific primers used for RT-qPCR are shown in Appendix A. Recombinase A (*recA*) was used as an internal gene standard for PCR amplification and data normalization. Normalized fold changes of the relative expression ratio between control and Cu(II) spiked samples were quantified by the 2^−ΔΔCT^ method [98]. All experiments were performed in triplicate using independent samples, and their mean value and standard error of the mean were calculated.

### 4.8. Network Analysis Using Cytoscape

The STRING database that integrates both known and predicted protein-protein interactions (PPIs) was used to predict functional interactions of proteins [99]. The PPI network between the significant DEGs (*p*-value < 0.05) was visualized using the PPI visualization software Cytoscape (version-3.9.0) [100]. DA-G20 was selected as the organism, and a confidence score cutoff of greater than 0.40 was used to find top interactors. Highly connected regions of the network were detected using ClusterONE (version 1.0) [101] algorithm with the following criteria: minimum size = 5, minimum density = 0.05, and edge weights = combined score. StringApp (Cytoscape plugin) was used to perform pathway enrichment analysis and import PPI networks from the STRING database to Cytoscape [102]. The most enriched gene set was selected based on a false discovery rate of 1.0 × 10^−6^. The most enriched pathway was selected for analyzing the interaction network, and a doughnut graph was assigned to each of the nodes with upregulated and downregulated log_2_FC values. In the PPI network, the nodes correspond to the proteins, and the edges represent the interactions.

### 4.9. Bioinformatics Analysis

Significantly differentially expressed genes (*p*-value < 0.05) were annotated with the respective proteins using the UniProt database [103]. The differentially expressed genes were further subcategorized based on their gene ontology terms (biological process, molecular functions, and cellular component) and pathways using the Quick GO database and BioCyc pathway/genome database collection [104,105]. R statistical programming language was used for computational analysis and generation of figures.

### 4.10. Targeted Metabolomics Analysis

Targeted metabolomics analysis was performed on 0 µM, 5 µM, and 15 µM Cu(II) by sending samples to the Metabolomics lab at the University of Illinois, Urbana Champaign, USA. The detailed procedure is outlined in the study by Gomez et al. [106]. Briefly, metabolites were extracted with 1 mL of 70% methanol and sonicated using an ultrasonic homogenizer. The lysed cell pellets thus obtained were fractionated with 70% methanol and chloroform and finally centrifuged for 10 min at maximum speed. One mL of each extract was evaporated under vacuum at −60 °C, and dried extracts were derivatized. Five microliters (5 μL) of the internal standard (hentriacontanoic acid (10 mg/mL); Sigma-Aldrich, St. Louis, MO, USA) was added to each sample prior to derivatization, and all the samples were analyzed on a GC/MS system (Agilent Technologies, Santa Clara, CA, USA) consisting of an Agilent 7890 gas chromatograph, an Agilent 5975 mass selective detector, and an HP 7683B autosampler.

All the chromatogram peaks (spectra) were compared with electron impact mass spectrum libraries NIST08 (NIST, Gaithersburg, MD, USA), W8N08 (Palisade Corporation, Ithaca, NY, USA), and a custom-built library of 520 unique metabolites [106]. The data obtained were normalized to the internal standard in each chromatogram and the sample dry weight (DW). AMDIS 2.71 (NIST, Gaithersburg, MD, USA) was used to evaluate the spectra of all chromatogram peaks. All metabolite concentrations are reported as relative concentration, i.e., ‘analyte concentration relative to hentriacontanoic acid per gram dry weight (for cell pellets) or per mL (for supernatant)’. More details on GC/MS data transformation are provided in the Appendix A.

## 5. Conclusions

The introduction of Cu(II) stress to DA-G20 caused common heavy-metal-associated oxidative stress as well as effects that seem to be exclusive during Cu(II) exposure. The most common effects observed in this study were: (i) energy conservation through translation arrest; (ii) downregulation of chemotaxis and motility; (iii) differential expression of certain transporter systems and inorganic ion transport complexes; (iv) upregulation of sigma factor proteins; and (v) the increased production of organic acids and amino acids. In addition, Cu(II) stress was found to disrupt the homeostasis of other heavy metals such as iron and zinc. Furthermore, DA-G20 was able to detoxify Cu(II) ions through four different strategies: (i) prevention of metal ions from entering the cells through the downregulation of several inorganic ion transporter complexes; (ii) the active transport of the metal ions out of the cell through the up-regulation of specific transporter complexes; (iii) intracellular sequestration of the Cu(II) ions by proteins rich in peptides with specific metal-binding amino acids; and (iv) through synthesis of specific metabolites that resulted in altering the cellular metal buffering system to prevent oxidative damage. The observed responses and differential gene expression in this study were primarily caused by the secondary effects of general heavy-metal toxicity. Some changes such as upregulation of ApbE family lipoprotein observed in this study seem to be specific for DA-G20 exposed to Cu(II) ions. However, further research is required to elucidate the underlying signaling processes involved in Cu(II) stress-response management. Our findings provide a framework for further research on the mechanism of Cu(II) toxicity in DA-G20. However, Cu(II) stress response seems to be a complex process that includes various regulatory processes. Integrated transcriptomic, metabolomics and proteomic analysis of DA-G20 will be necessary to validate the roles of identified genes during Cu(II) stress.

## Figures and Tables

**Figure 1 ijms-23-01396-f001:**
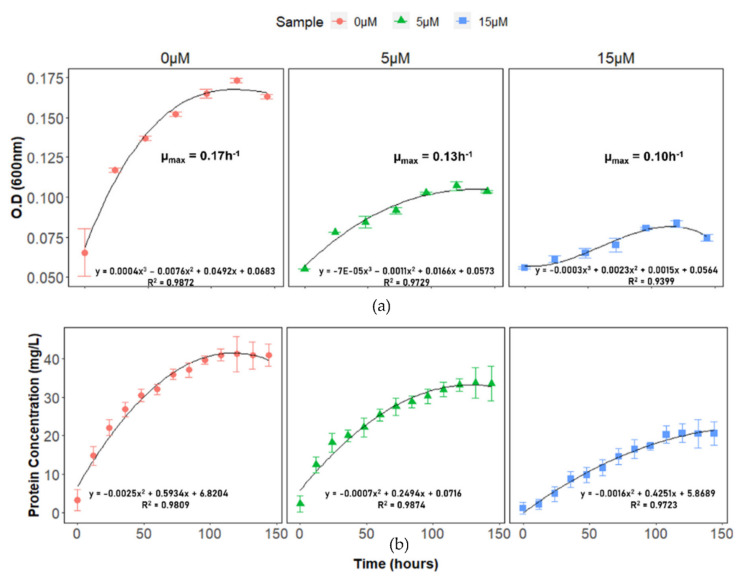
(**a**) Growth curve of DA-G20 in 0, 5, and 15 µM Cu(II); (**b**) Effect of Cu(II) on the growth DA-G20 as a function of total cell protein in presence of 0, 5 and 15 µM Cu(II). The points for all samples are the average of triplicates, and error bars indicate ±standard deviations of the means. Error bars smaller than the symbols are hidden behind sample legends. The best fit polynomial equation and goodness of fit (R^2^) is shown corresponding to each plot.

**Figure 2 ijms-23-01396-f002:**
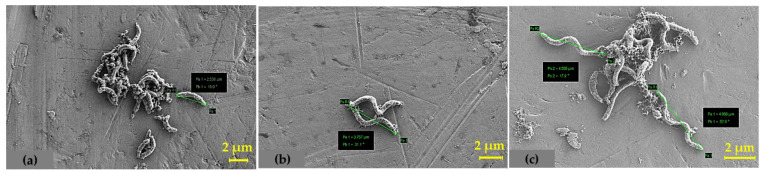
SEM image depicting variation in cell size of DA-G20 grown in media spiked with (**a**) 0 µM Cu(II), (**b**) 5 µM Cu(II), and (**c**) 15 µM Cu(II).

**Figure 3 ijms-23-01396-f003:**
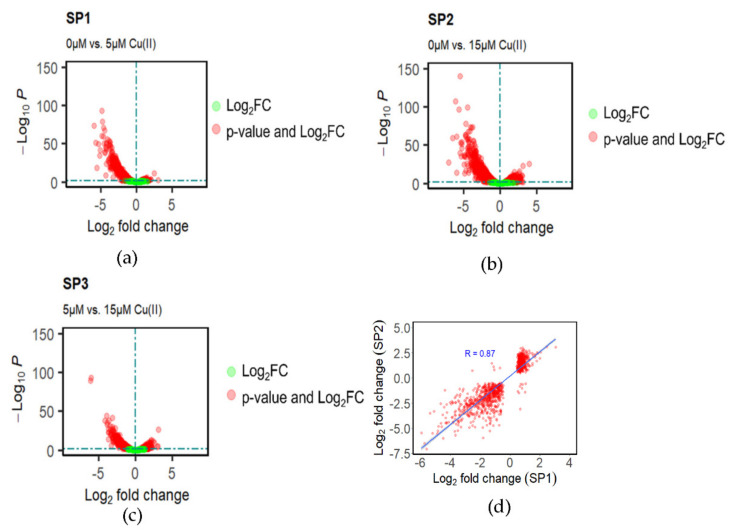
Differentially expressed genes in (**a**) SP1 (0 µM vs. 5 µM Cu(II)); (**b**) SP2 (0 µM vs. 15 µM Cu(II)); (**c**) SP3 (5 µM vs. 15 µM Cu(II)); (**d**) Correlation between common genes in SP1 (0 µM vs. 5 µM Cu(II)) and SP2 (0 µM vs. 15 µM Cu(II)). Red dots are all significant (*p*-value < 0.05) DEGs; Green dots are all insignificant genes with |log_2_FC| > 0.

**Figure 4 ijms-23-01396-f004:**
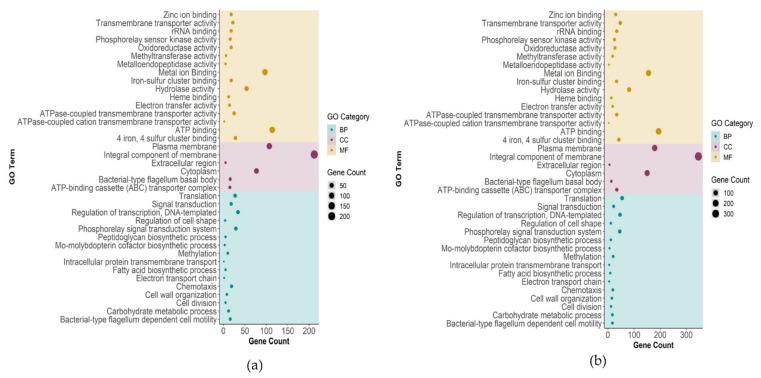
Bubble plot depicting gene counts of enriched GO terms in (**a**) 0 µM vs. 5 µM Cu(II); (**b**) 0 µM vs. 15 µM Cu(II). BP—Biological process; CC—Cellular component; MF—Molecular function.

**Figure 5 ijms-23-01396-f005:**
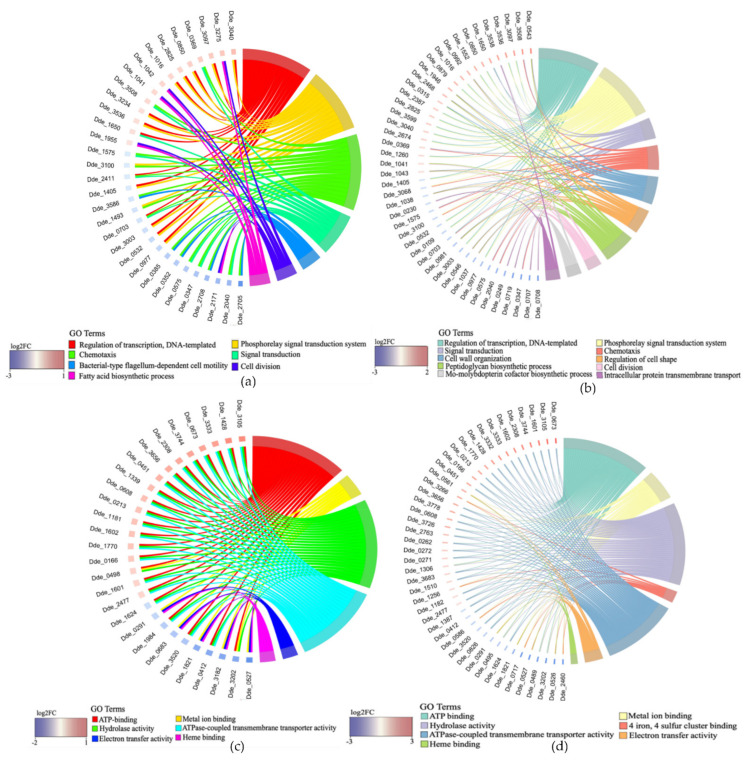
GOChord plot showing assignment of genes to their respective GO functional categories (BP and MF) according to their fold change in (**a**) SP1 (BP); (**b**) SP2 (BP); (**c**) SP1 (MF); (**d**) SP2 (MF). BP—Biological process; MF—Molecular function.

**Figure 6 ijms-23-01396-f006:**
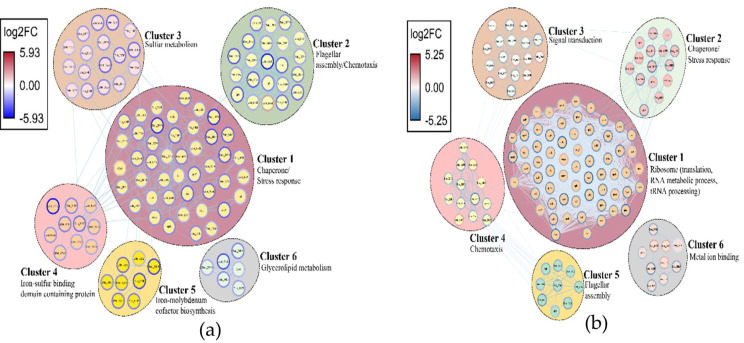
Protein-Protein interaction network in (**a**) 0 µM vs. 5 µM Cu(II) (SP1), and (**b**) 0 µM vs. 15 µM Cu(II) (SP2). Each cluster is a set of highly connected nodes and is illustrated in a circle.

**Figure 7 ijms-23-01396-f007:**
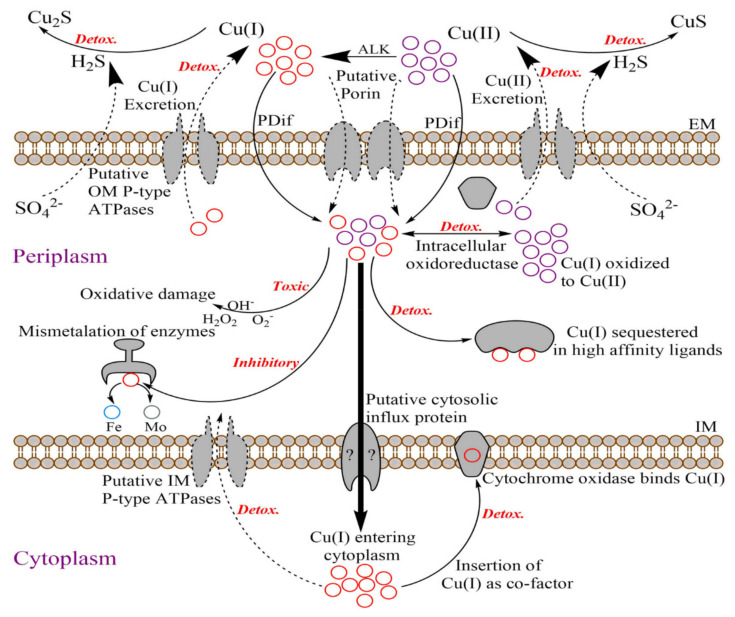
Schematic of putative copper transport and detoxification mechanism in *Desulfovibrio vulgaris* G20. ALK, Alkaline phosphatase; PDif, Passive diffusion; EM, Extracellular membrane; IM, Intracellular membrane; *Detox.,* Detoxification mechanism.

**Table 1 ijms-23-01396-t001:** Top 10 upregulated and downregulated genes in SP1 (0 µM vs. 5 µM Cu(II)) and SP2 (0 µM vs. 15 µM Cu(II)).

Gene ID	Protein Name	log_2_FC	Standard Error
Dde_2535 *	ApbE family lipoprotein	3.95	±0.37
Dde_2958 *	Flagellar basal body rod protein	3.12	±0.31
Dde_3047 *	Protein serine/threonine phosphatase	3.08	±1.19
Dde_1981 *	Uncharacterized protein	3.05	±0.93
Dde_0899 *	Uncharacterized protein	2.98	±0.48
Dde_1205 *	Uncharacterized protein	2.96	±0.44
Dde_2895 *	Teichoic-acid-transporting ATPase	2.85	±0.96
Dde_1929 *	Uncharacterized protein	2.82	±0.49
Dde_2799 *	Phage regulatory protein, Rha family	2.81	±0.63
Dde_0900 *	RNA polymerase sigma factor, sigma-70 family	2.77	±0.54
Dde_0111 ^#^	Zinc resistance-associated protein	−8.56	±0.30
Dde_4025 ^#^	Uncharacterized protein	−7.07	±0.64
Dde_2170 ^#^	UPF0235 protein Dde_2170	−6.48	±0.39
Dde_2819 ^#^	Uncharacterized protein	−6.11	±0.27
Dde_3737 ^#^	Putative GAF sensor protein	−5.98	±0.77
Dde_2991 ^#^	Transcription termination/antitermination protein, NusG	−5.97	±0.36
Dde_3226 ^#^	Phage shock protein A, PspA	−5.68	±0.27
Dde_1010 ^#^	Uncharacterized protein	−5.52	±0.22
Dde_0221 ^#^	Response regulator receiver protein	−5.51	±0.41
Dde_0715 ^#^	Uncharacterized protein	−5.46	±0.35
Dde_3047 **	Protein serine/threonine phosphatase	3.05	±1.03
Dde_2958 **	Flagellar basal body rod protein	2.48	±0.36
Dde_0959 **	AIG2 family protein	2.04	±0.60
Dde_3729 **	ABC transporter related protein	2.01	±0.55
Dde_4035 **	Uncharacterized protein	1.87	±0.60
Dde_1264 **	PAS modulated sigma54 specific transcriptional regulator, Fis family	1.82	±0.67
Dde_3378 **	Uncharacterized protein	1.80	±0.61
Dde_0930 **	Uncharacterized protein	1.70	±0.53
Dde_3061 **	M18 family aminopeptidase	1.68	±0.69
Dde_0925 **	Uncharacterized protein	1.67	±0.62
Dde_0715 ^##^	Uncharacterized protein	−5.93	±0.32
Dde_2170 ^##^	UPF0235 protein Dde_2170	−5.72	±0.37
Dde_4025 ^##^	Uncharacterized protein	−5.58	±0.63
Dde_0356 ^##^	Flagellar basal body rod protein FlgB	−5.34	±0.35
Dde_0221 ^##^	Response regulator receiver protein	−5.17	±0.42
Dde_0563 ^##^	Uncharacterized protein	−5.10	±0.37
Dde_1010 ^##^	Uncharacterized protein	−4.88	±0.23
Dde_2560 ^##^	Thioredoxin peroxidase	−4.84	±0.29
Dde_1689 ^##^	OmpA/MotB domain protein	−4.78	±0.25
Dde_0283 ^##^	Uncharacterized protein	−4.64	±0.26

*—Top 10 upregulated genes in 0 vs. 15 µM Cu; ^#^—Top 10 downregulated genes in 0 vs. 15 µM Cu(II); **—Top 10 upregulated genes in 0 vs. 5 µM Cu; ^##^—Top 10 downregulated genes in 0 vs. 5 µM Cu(II). All the protein annotations for the genes were retrieved from UniProt database.

**Table 2 ijms-23-01396-t002:** The top five most enriched GO terms with their respective *p*-values and z-score in both SP1 and SP2.

Enriched GO Terms in SP1 (0 vs. 5 µM Cu)				
Gene Ontology (GO) Term	GO ID	Total Gene Count	−Log_10_ (*p*-Value)	z-Score *
Regulation of transcription, DNA-templated (BP)	GO:0006355	34	16.5	0
Phosphorelay signal transduction system (BP)	GO:0000160	29	2.60	−1.29
Translation (BP)	GO:0006412	27	48.14	−4.81
Chemotaxis (BP)	GO:0006935	19	2.74	−2.98
Signal transduction (BP)	GO:0007165	18	2.73	−1.88
ATP binding (MF)	GO:0005524	114	16.55	2.06
Metal ion binding (MF)	GO:0046872	97	2.97	−2.94
Hydrolase activity (MF)	GO:0016787	54	1.96	0
4 iron, 4 sulfur cluster binding (MF)	GO:0051539	28	2.32	−0.75
ATPase-coupled transmembrane transporter activity (MF)	GO:0042626	25	1.64	3.40
Integral component of membrane (CC)	GO:0016021	212	3.38	−0.41
Plasma membrane (CC)	GO:0005886	99	1.94	−0.90
Cytoplasm (CC)	GO:0005737	76	1.77	−2.52
ATP-binding cassette (ABC) transporter complex (CC)	GO:0043190	15	1.58	2.32
Bacterial-type flagellum basal body (CC)	GO:0009425	9	5.05	−3.00
Enriched GO terms in SP2 (0 vs. 15 µM Cu)				
Translation (BP)	GO:0006412	54	51.75	−7.07
Regulation of transcription, DNA-templated (BP)	GO:0006355	45	3.59	1.5
Phosphorelay signal transduction system (BP)	GO:0000160	45	2.30	0.15
Signal transduction (BP)	GO:0007165	22	5.76	−0.85
Methylation (BP)	GO:0032259	20	4.26	0
ATP binding (MF)	GO:0005524	194	72.25	1.29
Metal ion binding (MF)	GO:0046872	152	3.30	−1.13
Hydrolase activity (MF)	GO:0016787	81	4.37	1.22
Transmembrane transporter activity (MF)	GO:0022857	47	3.20	4.23
4 iron, 4 sulfur cluster binding (MF)	GO:0051539	42	2.33	0.3
Integral component of membrane (CC)	GO:0016021	346	17.47	4.3
Plasma membrane (CC)	GO:0005886	160	1.66	1.42
Cytoplasm (CC)	GO:0005737	149	1.52	−5.65
ATP-binding cassette (ABC) transporter complex (CC)	GO:0043190	34	1.73	4.11
Bacterial-type flagellum basal body (CC)	GO:0009425	6	2.76	−0.81

BP: Biological Process; MF: Molecular Function; CC: Cellular component. * z-score = $\frac{\left(\mathrm{Total}\;\mathrm{number}\;\mathrm{of}\;\mathrm{upregulated}\;\mathrm{gene}-\mathrm{Total}\;\mathrm{no}.\;\mathrm{of}\;\mathrm{downregulated}\;\mathrm{gene}\right)}{\mathrm{Total}\;\mathrm{gene}\;\mathrm{count}}.$ Note: This z-score does not refer to the standard score from statistics. It calculates if the GO term is more likely to be decreased (negative value) or increased (positive value) in expression.

## Data Availability

The sequences were uploaded and deposited in the National Center for Biotechnology Information (NCBI) Gene Expression Omnibus (GEO) under the accession number GSE191076.

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
