# Peer review of "Transcriptomics and Functional Analysis of Copper Stress Response in the Sulfate-Reducing Bacterium Desulfovibrio alaskensis G20"

_ijms, 2022, doi:10.3390/ijms23031396_

Round 1

Reviewer 1 Report

This work has been very thoroughly and substantively prepared and I have no significant comments. The paper brings a dose of new knowledge in the field of biology and biochemistry, which is necessary for the protection of the environment and and at the same time - the human health.

Comments:

- In my opinion, in publications of such a high level, reporting of research results should only be in an impersonal form. I do not accept the first person in scientific papers. As I guess, the authors wanted to emphasize the original character of the scientific achievement in this way.

- Moreover - it is more logical include - as the first chapter "Material and Method", and then "Results"  (research was planned, prepared and realized first).

- Clearly emphasize in the last paragraph of the Introduction the purpose of your work and the way to achieve it, please.

My comments will not change the fact that the article is interesting, scientifically valuable and should be published as soon as possible.

Reviewer 2 Report

Copper stress in Desulfovibrio alaskensis G20 is an interesting topic for research and the molecular approach has provided some useful information.     

There is a correction to be made in line 107 and 114. Change “figure 1A” to “Figure 1a” and “figure 1B” to “Figure 1b”.

As mentioned in line 106,  the presence of Cu(II) resulted in a change in cell morphology. However, there are no additional statements about how the cell morphology has changed. The legend for Figure 1S indicates a change in  cell size and more discussion (including measurements, etc.) is needed on this topic. In the Figure 5S, there was no change in regulation of genes for cell shape and the genes for cell division were upregulated in the presence of Cu(II). Is the change in cell size (length?) attributed to protein (end enzyme inhibition) ?

The authors consider change in cell morphology  as an important feature because they have it in the first sentence of the results. And, in the abstracts there is a mention of copper “generaging pleiotropic effects in the microbial cell” which could refer to the change in cell size. Perhaps Figure !S should be moved into the results section and out of the supplemental material.
